# Investigating the Relationships between Taste Preferences and Beverage Intake in Preadolescents

**DOI:** 10.3390/foods12081641

**Published:** 2023-04-14

**Authors:** Eva Winzer, Marlies Wallner, Anna Lena Aufschnaiter, Daniela Grach, Christina Lampl, Manuel Schätzer, Barbara Holstein, Maria Wakolbinger

**Affiliations:** 1Centre for Public Health, Department of Social and Preventive Medicine, Medical University of Vienna, 1090 Vienna, Austria; 2Institute of Dietetics and Nutrition, University of Applied Sciences FH JOANNEUM, 8020 Graz, Austria; 3Competence Center Climate and Health, Austrian National Institute of Public Health, Stubenring 6, 1010 Vienna, Austria; 4Special Institute for Preventive Cardiology and Nutrition—SIPCAN, 5061 Salzburg, Austria

**Keywords:** taste perception, sweet intensity rating, sweet liking, PROP, adolescents, anthropometry, beverage intake

## Abstract

Sugar-sweetened beverages are known promotors of adverse health outcomes. This study aimed to find a relation between taste perception, preferences for beverages, anthropometric parameters, and frequency of beverage consumption. Taste perception of sweetness was tested using an adopted sensitivity test with sucrose and different concentrations of sugar-sweetened apple juice. Furthermore, bitter-compound 6-n-propylthiouracil (PROP) and salty perception were tested and accompanied by a questionnaire on beverage intake. We did not find a clear relationship between taste perception, anthropometrics, and beverage intake. Nevertheless, in males, the bitter intensity perception of PROP was positively correlated with the BMI percentiles (CDC, r = 0.306, *p* ≤ 0.043) and the waist circumference (r = 0.326, *p* = 0.031). Furthermore, the liking of sweet taste (*p* < 0.05) and sweet intensity rating (*p* < 0.05) of apple juice increased with intensity, and adolescents with overweight or obesity had a higher intake of free sugars from beverages (*p* < 0.001). The role of taste perception on anthropometric measures and beverage intake remains unclear and requires further investigation.

## 1. Introduction

Looking at various European countries, nearly one in five 15-year-olds suffer from being overweight or obese, with boys being at higher risk than girls. On average, 23% of boys and 15% of girls are living with overweight or obesity at 15 years of age [1]. Considering this worldwide health problem, sugar-sweetened beverages (SSBs) play, among other factors, a leading role. Numerous studies demonstrate clear associations between SSB consumption and a change in body weight measures [2]. Adolescents who increased their SSB intake showed more body fat and waist circumference after two years [3]. A study with over 2000 children aged 7 to 18 years showed that the consumption of more than 120 mL of SSBs per day is clearly associated with an increased BMI and an increased risk for abdominal obesity [4], which was similarly shown in cohort studies in children [5]. Moreover, a recent meta-analysis including over 120,000 children and adolescents revealed that a high SSB intake was significantly associated with higher waist circumference and a higher percentage of body fat [6].

Increasing body fat mass is a consequence of regular SSB consumption and a higher risk of type 2 diabetes mellitus, cardiovascular diseases, some cancers, and dental problems, such as caries, are also well-studied SSB-related health problems [7]. For instance, cholesterol levels were shown to change significantly in children and adolescents when drinking SSBs regularly [8].

Regarding preferences for SSBs, in a European study (IDEFICS study), primary school children preferring sugar-sweetened juice had 50% higher odds of living as overweight or obese individuals than children preferring natural juice. It was shown that children following a dietary pattern with a higher sugar and fat intake have a 17% increased risk of gaining excessive weight two years later [9]. Similar results were found in a study in Poland with 150 children aged eight to 15 years: children with a high sweet taste preference were twice as likely to suffer from overweight or obesity compared to children with a lower sweet taste preference [10]. In another European study including eight different countries with almost 1700 children between six and nine years of age, both overweight and obesity were positively associated with sweet taste preference when comparing natural versus sugar-sweetened apple juice. Furthermore, when a combination of sweet taste and fat taste preference is present, an exceptionally high probability of obesity could be detected, but only in girls [11]. Therefore, sensory behaviors are associated with children’s consumption of SSBs [12].

In addition, the ability to taste bitter and its link to overweight and obesity has recently been the subject of research, but results have been partly inconclusive. Generally, it is known that children who have high responsiveness not only to sweet but also to salty, umami, or sour taste tend to respond more to bitter as well [13]. Children who cannot taste 6-n-propylthiouracil (PROP), called non-tasters, were found to have higher BMI z-scores than tasters. Similar results were demonstrated in another study in which children with overweight and/or obesity were more likely to be sensitive to bitter taste perception. Especially girls have a higher probability of belonging to this group [14]. A study investigating the relationship between PROP taster status and dietary patterns among children showed that processed foods were positively associated with body composition for the non-tasters but not the tasters, indicating that PROP taster status can moderate the relationship between food consumption patterns and body composition. Nevertheless, taster status did not contribute to the prevalence of overweight or obesity in this study [15]. The same applies to adults, showing an association between gustatory sensitivities and macronutrient intake and choice in general [16]. In contrast, other studies with children showed significantly more tasters in the underweight category than in the overweight category [17]. However, in adolescents, no significant correlation between overweight or obesity and PROP-taster status could be demonstrated [18]. Furthermore, other dietary patterns among tasters and non-taster can be observed, e.g., the liking of carrots and bread among children with low sensitivity to bitter [19]. In the Italian Taste project, it was shown among adults that being a taster was negatively associated with a liking for rockets and radishes [20].

Based on these findings, we assumed that taste perception and preferences for beverages are associated with the intake of SSBs and anthropometric markers in 11- to 12-year-old adolescents.

## 2. Materials and Methods

### 2.1. Participants

In this study, 118 sixth-grade children (69% 11-year-old and 31% 12-year-old; 49% girls and 51% boys) were included from two public secondary schools in Styria, Austria. One school was from a rural, and the other one was from an urban area. Information on the study was provided to preadolescents and their parents. To participate in the study, preadolescents and their parents had to sign a written consent form. A total of 108 preadolescents returned the consent form. Eight participants did not participate in the sensory testing and/or anthropometry measurements or did not return the questionnaires, resulting in 100 preadolescents involved in the data analyses (Figure 1). The study was approved by the Regional Educational Authorities. The ethical approval of this study has been granted by the ethics committee of the Medical University of Graz No. 31-439 ex 18/19, and while the General Data Protection Regulation (GDPR) has been followed for data protection, it makes reference to the Declaration of Helsinki.

### 2.2. Procedure

In the beginning, the study team explained the procedure and rules (i.e., preadolescents should not talk to each other; they should rinse their mouths with water in-between samples). The preadolescents completed the sensory testing, anthropometry measurements, and the questionnaire in the preadolescent’s respective schools and classrooms from 8 to 12 a.m. The study team was present during the testing time, guided the preadolescents through the questionnaire, and explained every question to avoid misunderstandings and mistakes.

The questionnaire consisted of sociodemographic data; thus, age, sex, and migration background (country of birth of mother and father) were assessed. The Family Affluence Scale II (FAS) [21] was used to determine socioeconomic status. The scale is composed of four questions, the number of cars and computers in the household, the availability of a separate preadolescent’s room, and the number of vacations taken by the family in the last calendar year [22].

Body height was determined with a tape measure while the preadolescents were standing without shoes and with feet together. Body weight was measured with preadolescents lightly clothed and also without shoes using a mobile calibrated scale, allowing body mass index (BMI) to be calculated. The sex- and age-specific cut-off values from the World Health Organization (WHO) [23] and the Centers for Disease Control and Prevention (CDC) [24] were used to estimate the prevalence of overweight and obesity. Waist circumference was measured with a tape under standardized procedures as instructed by the WHO (midpoint between the lower costal margin and iliac crest) [25]. 

Sensory testing aimed to measure the preadolescent’s responsiveness and liking to sweet tastes, the preadolescent’s sweet tastes detection (DT) and recognition (RT) thresholds, and the preadolescent’s responsiveness to PROP (bitter sensitivity) and salty sensitivity. All measurements were conducted by the same trained researchers within one day.

### 2.3. Samples

The sweet concentration level was adopted by Knof et al. [26]. These concentrations have been used to measure taste sensitivity in children between 6 and 9 years in a big population survey in Europe [9].

The participants’ sweet taste sensitivity was evaluated based on five concentration levels (Table 1). Just before the test, the samples were made by dissolving the stimuli in tap water. Exactly 10 mL of the sample solutions were served to the preadolescents at room temperature, and the same water was provided for neutralization.

### 2.4. Detection and Recognition Test (Sucrose Solution)

Detection (DT) and recognition thresholds (RT) were used to gauge the preadolescent’s sensitivity to sweet taste. Five water-based solutions (Table 1) were given to the preadolescents at the same time in small cups (10 mL servings), in addition to a large water cup clearly identified to use for rinsing and as a reference. The preadolescents were told to rinse the complete contents of, i.e., cup 0 and spit out the samples in the spitting cup and to carry out the sampling in a staircase sequence (i.e., from cup 0 to cup 4). They had to find out which cups contained pure water and which would taste different from pure water. So, they were told to associate the five taste samples presented with different symbols. Zero represented the answer, “I taste nothing, the sample tastes like water”, a question mark, “It tastes different from 0, but I can’t tell what it tastes like”, and X, “I have detected some kind of taste”. When the preadolescents had tasted all cups, they had to identify the taste. Five taste box options were offered: “bitter”, “sour”, “salty”, “sweet”, and “savoury”. The taste options were explained to the preadolescents with food examples, e.g., savory, like the taste of the soup.

We hypothesized that preadolescents who responded “water” could not perceive any taste at each concentration level (tastant under detection threshold). When the subject began to distinguish the sample from water, DT was obtained, and RT was acquired when the person accurately identified the taste. Finally, we presumed that preadolescents who responded “I don’t know” or incorrectly identified the taste quality could identify the tastant, so this level was recorded as their DT [27]. 

### 2.5. Responsiveness of Sweet Taste (Apple Juice)

The preadolescent’s taste intensity rating was combined with the liking of sweet taste [28]. Testing was performed with a natural food product (apple juice). The preadolescent’s responsiveness and liking of sweet taste were measured with five concentration levels (Table 1), and responses were recorded with a 5-point Likert scale from “not sweet” to “incredibly sweet”. At the same concentration level, the preference for sweet taste was also recorded on a 5-point Likert scale ranging from “not good” to “incredibly good”. According to Ervina et al., both were scaled to 0–100 for data processing reasons [27].

### 2.6. PROP and Salt Responsiveness

PROP (6-n-propylthiouracil) and salt intensity rating were measured by paper strips [29,30]. The bitter-tasting solution of 50 mM PROP and salty control solution (1 M NaCl) was applied on tasting strips by soaking, according to Zhao et al. [31]. The preadolescents were introduced to using the scale and the taste strips to ensure correct use. Both paper strips were consecutively placed (salty one first) in the preadolescent’s mouth and held there for approximately 5–10 s. Then, the taste perception, as well as the bitterness intensity, had to be rated using the general labeled magnitude scale (gLMS) [32] from “not detectable” to “strongest imaginable”. 

Participants were categorized as non-tasters (if their gLMS grade for bitterness (PROP solution) was ≤13 mm), medium-tasters (14–67 mm), and supertasters (>67 mm) [33]. To avoid demotivating supertasters from further involvement, the test was scheduled at the end of the entire testing session, or if that was not feasible, they obtained water for neutralization and had to take a break before the subsequent testing.

### 2.7. Beverage Intake Questionnaire

Patterns on beverage intake were assessed with the BEVQ-15 (15-item beverage intake questionnaire) [34]. The questionnaire comprises 15 beverage categories and an additional category for beverages not included in the other categories. Each category has the option to indicate the frequency and quantity of a particular beverage. To estimate the quantity better, in addition to information on milliliters, a diagram with common drinking vessels and their capacity was handed out. The beverages consumed were analyzed according to the DEBInet nutrition database [35] to determine free sugar intake per day.

### 2.8. Statistical Analysis

Statistical significance tests such as the student’s *t*-test or Mann–Whitney U test, ANOVA, and Chi-square test were used to compare the groups. Tukey’s post hoc test with a significance threshold of α = 0.05 was used to make pairwise comparisons between preadolescents’ sweetness sensitivity (DT and RT), responsiveness, and liking of sweet taste solutions were conducted using. Pearson’s or Spearman’s correlations were computed between the different sensitivity measurements depending on data distribution.

The statistical significance level is set at *p* < 0.05. All statistical analyses were performed using SPSS 27.0 (SPSS Inc., Chicago, IL, USA).

## 3. Results

### 3.1. Participants’ Characteristics

Table 2 summarizes the participants’ characteristics. Forty-nine percent of the participants were girls. Sixty-nine percent of the preadolescents were 11-year-olds (mean age = 11.3 years). Thirty-one percent of the participants had a migration background. Sixty-five percent of the preadolescents had families with high affluence, which means higher socioeconomic status. Forty percent of the participants were living as overweight or obese individuals (mean BMI = 20.1 kg/m^2^) and had a mean waist circumference of 72.1 cm. 

### 3.2. Preadolescent’s Sweetness Taste Sensitivity and Liking of Water Solutions

Table 3 summarizes the preadolescent’s sweetness sensitivity and liking, as measured by the different methods. The sweet taste showed to have a detection threshold (DT) level of 1.94 g/L sucrose and a recognition (RT) level of 5.87 g/L sucrose, meaning, on average, sweet was detected at this concentration.

The intensity rating on a gLMS scale by the preadolescents showed that they were partly able to distinguish between increasing concentrations, although only in every two steps. Meaning that lower sweet intensity ratings of 5 and 7 g sucrose/100 mL apple juices were reported compared to the two following higher concentrations (10 and 15 g sucrose/100 mL juice) and those again to the highest concentration (20 g sucrose/100 mL, *p* ≤ 0.001). 

There was a difference in the liking ratings of the preadolescents for the preference of all juices above 10 g sucrose/100 mL juice receiving higher ratings compared to the two lower concentrated juice samples (5 and 7 g sucrose/100 mL). 

Furthermore, the liking for the two lowest sucrose concentrations was significantly lower than all other tested concentrations from 10 g sucrose/100 mL and above.

There was a significant difference in the liking of sweet taste apple juice solution of 20 g sugar/100 mL between the age groups, with 11-year-olds preferring sweetness more than 12-year-olds (*p* = 0.036). There was a trend in the difference in liking the sweet taste of apple juice solution of 20 g sugar/100 mL depending on weight categories (*p* = 0.090). 

Based on Pearson correlation coefficients (r), there was a significant negative correlation between RT scores and salty responsiveness (r = −0.236, *p* = 0.041). Sweetness responsiveness level 5 (apple juice 20 g sugar/100 mL) showed a significant positive correlation with sweetness liking level 5 (r = 0.228, *p* = 0.028). Preadolescent’s sweetness sensitivity (DT and RT), responsiveness, and liking of sweet taste solutions according to BMI categories showed no significant differences except for liking sweet taste solutions. There was evidence that preadolescents with overweight or obesity had a higher mean liking of 20 g sugars/100 mL (apple juice) of 72.3 (28.7) mm compared to preadolescents with normal weight of 61.2 (29.4) mm, but not significantly different (*p* = 0.090). Underpinning this result, it was revealed that the liking for the most sugared apple juices was significantly associated with BMI (r = 0.211, *p* = 0.042) but not with BMI percentiles.

### 3.3. Preadolescent’s Overall Taste Perception (PROP and Saltiness)

The classification of preadolescents to PROP taster status based on the intensity rating of bitterness resulted in 13% non-tasters, 24% medium-tasters, and 63% supertasters.

Responsiveness to salty tastes showed a significant positive correlation with responsiveness to bitter (r = 0.507, *p* < 0.001). The sweet sensitivity measurements were not significantly correlated to bitterness and saltiness testing results. Except that, with a lower recognition level for sweet taste, a higher intensity rating for the salty taste strip was detected (*p* = −0.236, *p* = 0.041). In the overall group, no significant correlation between PROP intensity scoring and any BMI—percentiles (CDC, WHO, or z-Score) nor waist circumference was found. Further analysis revealed that in males, the bitter intensity perception of PROP was positively correlated with the BMI percentiles (CDC, r = 0.306, *p* = 0.043) and waist circumference (r = 0.326, *p* = 0.031), meaning a higher intensity perception of the bitter taste with increasing BMI and waist circumference. There was no correlation between females and saltiness. Furthermore, we did not find any significant differences in the PROP bitterness nor the salty ratings between males and females in the overall group.

### 3.4. Preadolescent’s Beverage Intake

Based on the beverage intake questionnaire (BEVQ-15), the mean liquid consumption was 1966 (SD = 1303) mL per day, and the mean sugar consumption was 55.6 (73.5) g per day. By using the WHO recommendations for sugar intake [36], 37% of the preadolescents consumed more than 10% and 52% more than 5% of total energy intake from free sugars from drinks (mean = 9.5%).

The preadolescent’s beverage intake according to BMI categories is shown in Figure 2. There was a significant difference between participants with normal weight and preadolescents living with overweight or obesity regarding free sugars from beverage intake (*p* < 0.001) and the percentage of total energy intake from sugars (*p* = 0.013).

## 4. Discussion

The aim of the present study was to examine the relationship between taste sensitivity, acceptance, intake of SSB, and anthropometric measures in Austrian preadolescents. Out of the 100 participating preadolescents (11–12 years), 40% were living with overweight or obesity, categorized by BMI for age percentiles. This is a very high proportion compared to other investigations, which reported 19% [37] and 31% [38] of children living with overweight or obesity. The children included in the other examinations were partly younger (9 to 12 years of age), and the prevalence might be lower in younger ones here. However, this might only partially account for the high prevalence.

Overall, we found no clear relationship between sensory testing results and anthropometric markers. Nevertheless, adolescents with overweight or obesity had a significantly higher consumption of free sugars from beverages and the percentage of total energy from sugars. These results are confirmed by several investigations showing sugary beverages are more likely to be associated with the development of overweight or obesity and negative changes in anthropometric parameters in children and adolescents [2,3,4,5,6]. 

To our knowledge, the PROP-taster status is reported for the first time in Austrian school preadolescents. In Austria, 87% of participating preadolescents were allocated to tasters or supertasters. Compared to other studies dealing with younger children, this is a rather high proportion. In other studies, 40.4% of the girls and 48.9% of the boys were identified as tasters, or only 41% of tasters in a study conducted in New Zealand [15], although, in an Italian study with adults, 77% were tasters [39].

We did not observe any effects of the taster status or the responsiveness to bitter with other perceptions, except for salt perception (r = 0.507, *p* < 0.001), which was tested as a control substance in the entire group. In contrast, there is one study with adults showing that PROP bitterness ratings were correlated positively to the intensity of different flavors [40]. A more detailed analysis showed a weak association between males’ increasing BMI for age percentiles (r = 0.191, *p* = 0.033) and their perception of PROP-bitterness. Here, it can be hypothesized that with increasing sensitivity, more healthy food items are more likely to be rejected, such as bitter-tasting vegetables, especially green ones, compared to individuals who perceive the bitter compound PROP as less intensive. This is supported by a review [41], and a recent study that found a lower preference for bitter-tasting food and the presence of the (C/G) variant of the TAS2R38 gene (bitter-tasting receptor) were associated with the development of obesity [42]. Unhealthy foods are consumed more, although we did not find any correlation between PROP intensity rating and the intake of energy, sugar, or fat in males. We know that this result should be interpreted carefully and suggest this be considered in future studies. Especially concerning other study results, higher acceptance of high-fat food in non-tasters (perceive bitter less or not intense [43]) and lower BMI in young males [29,44] are reported. However, the focus of this study was on beverage intake, so we cannot confirm this hypothesis. With regards to PROP bitterness ratings, the Italian Taste study (*n* = 1225) showed a significant difference between male and female adults [20]. In our study, male and female schoolchildren showed similar ratings. Although the age of our study population was much lower and the number of participants, which might have limited the significance.

In the present study, the sweet intensity rating of the different sucrose concentrations was feasible (Table 3). It was not possible to distinguish between the two lowest (5 and 7 g sucrose/100 mL) and between the two middle (10 and 15 g sucrose/100 mL) concentrated samples. However, the intensity rating increased with higher concentrations. Furthermore, the liking of the samples with 10 g sucrose and more was significantly greater. Overall, these results suggest that the participating preadolescents were able to rank sugar concentrations and showed a stronger liking for higher sweet perceptions.

Furthermore, we did not find a clear relationship between the liking for the sweetness of juices (even not with a higher concentration of sugars) and the existence of overweight or obesity. However, we identified a non-significant weak relationship (*p* < 0.1) that the juice with extremely high sugar content (20 g sucrose/100 mL) was liked differently across weight groups. These unclear results need further clarification as in a Polish study, it was revealed that kids who preferred the sweeter sample lived with obesity more frequently than kids who showed normal weight [10].

Furthermore, the IDEFICS study [11] revealed that the preference for the sweeter juice was associated with being overweight, although only in girls and in combination with a preference for a fatty taste. In relation to these findings, we found that the group with overweight consumed more free sugars and energy from beverages (Figure 1), which may have an impact on their liking for sweets.

Regarding sweet taste threshold measurements, we did not find any association between the BMI percentiles and the detection or recognition levels. This is comparable to other research that did not find associations between taste perception thresholds and BMI or overweight in Norwegian preadolescents (mean age 11 years [45]) nor in Mexican adults [46]. Although in the study of Ervina et al. [45], a Principal Component Analysis revealed positive associations between higher detection thresholds in different taste stimuli (including bitter) and BMI levels.

The detection threshold in the study of Petty et al. [47] was a bit lower for children (<10 years, ~10 mM) compared to our results (~14 mM). Both of these studies conducted with children seem to have higher thresholds (=lower sensitivity) compared to adolescents (10–19 years, ~8 mM) and adults (~7 mM), reflecting a change in sensitivity of perception during development.

Although the study was planned thoroughly, we must deal with the limitation that the number of participants is at the lower range for sensory testing. While we had planned on more preadolescents, we were not able to receive all results due to illness or missing informed consent. Nevertheless, the first results for Austrian adolescents regarding sensory perception and liking of sugar and bitter compounds are reported, despite the unclear outcomes.

## 5. Conclusions

As this study aimed to assess the association between taste perception, preferences for beverages, anthropometric parameters, and frequency of beverage consumption, we found that, in male preadolescents, the bitter intensity perception of PROP was positively correlated with the BMI percentiles and waist circumference. Moreover, the liking of sweet taste and sweet intensity rating of apple juice samples increased with intensity. Finally, preadolescents with overweight or obesity had a higher intake of free sugars and energy from beverages. Further research is needed to evaluate the role of sensory perception on metabolic conditions and sugar-sweetened beverage intake in children as a known promotor of adverse health outcomes. In addition, the high prevalence of overweight and obesity demonstrates the need for urgent actions in this age group.

## Figures and Tables

**Figure 1 foods-12-01641-f001:**
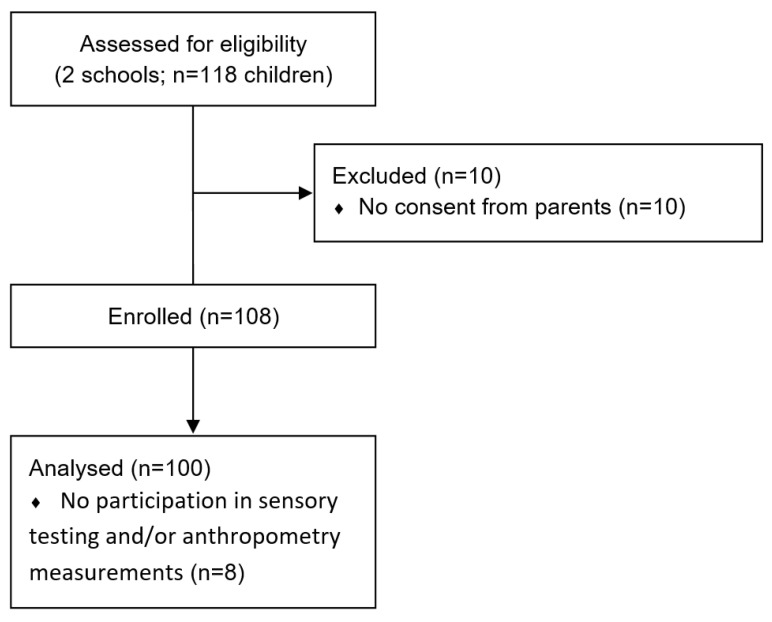
Flow chart of participant recruitment.

**Figure 2 foods-12-01641-f002:**
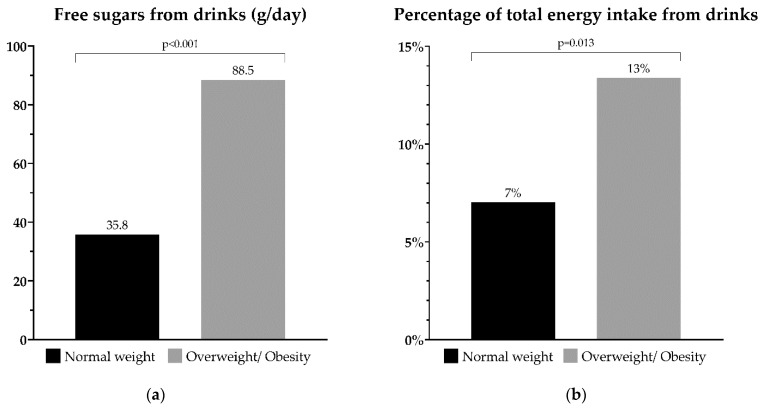
Preadolescent’s beverage intake according to BMI categories. (**a**) Sugars in g/day of beverage intake; (**b**) Percentage of daily energy intake from sugars of beverage intake.

**Table 1 foods-12-01641-t001:** Sweet taste sensitivity and responsiveness and liking of sweet taste.

	Taste Compound		Level 1	Level 2	Level 3	Level 4	Level 5
Sweet taste sensitivity Detection (DT) and recognition thresholds (RT)	Sucrose	mmol/L	0	5.8	14.6	26.3	38.0
		g/L	0	2	5	9	13
		g/100 mL	0	0.2	0.5	0.9	1.3
Responsiveness and liking of sweet taste	Apple juice	g/100 mL	5	7	10	15	20

**Table 2 foods-12-01641-t002:** The participants’ characteristics.

	Participants (*n* = 100) *n* (%) or Mean (SD)
Sex	
Boy	51 (51%)
Girl	49 (49%)
Age (years)	11.3 (0.5)
11-year-old	69 (69%)
12-year-old	31 (31%)
Migration background	
Natives	69 (69%)
Native-born with a migration background	21 (21%)
Foreign-born with a migration background	10 (10%)
Family Affluence Scale (FAS)	
Low affluence	8 (8%)
Middle affluence	27 (27%)
High affluence	64 (65%)
BMI (kg/m^2^)	20.1 (5.0)
Underweight	13 (13%)
Normal weight	45 (47%)
Overweight	18 (19%)
Obesity	20 (21%)
BMI for age percentile (CDC)	59.0 (33.8)
BMI for age percentile (WHO)	60.4 (34.3)
BMI for age z-score (WHO)	0.5 (1.6)
Waist circumference (cm)	72.1 (13.5)

BMI = body mass index; CDC = Centers for Disease Control and Prevention; WHO = World Health Organization.

**Table 3 foods-12-01641-t003:** Preadolescent’s sensitivity (DT and RT), intensity rating, and liking of sweetness.

Sweetness	Mean (SD)
Detection (DT) of sugar solutions	
Level ^1^	0.78 (1.01)
Concentration, g/L ^2^	1.94 (3.02)
Recognition (RT) of sugar solutions	
Level ^1^	2.11 (1.09)
Concentration, g/L ^2^	5.87 (3.6)
Sweetness intensity rating of apple juice ^3^	
5 g sugars/100 mL, mm	50.27 (28.28) ^a^
7 g sugars/100 mL, mm	48.64 (31.31) ^a^
10 g sugars/100 mL, mm	56.45 (29.23) ^b^
15 g sugars/100 mL, mm	60.37 (29.88) ^b^
20 g sugars/100 mL, mm	69.95 (30.58) ^c^
Liking of apple juice ^4^	
5 g sugars/100 mL, mm	59.07 (29.93) ^a^
7 g sugars/100 mL, mm	56.79 (29.67) ^a^
10 g sugars/100 mL, mm	64.95 (26.48) ^b^
15 g sugars/100 mL, mm	68.48 (28.44) ^b^
20 g sugars/100 mL, mm	65.59 (31.04) ^b^

^1^ Mean levels 1–5. ^2^ Mean concentrations in g/L determined from concentrations that match the mean level. ^3^ Mean responsiveness was measured on sweet taste apple juice solution of level 1–5 concentration (Table 1). ^4^ Mean liking was measured on sweet taste apple juice solution of level 1–5 concentration (Table 1). ^a–c^ Tukey’s test revealed a significant difference at *p* < 0.05.

## Data Availability

The data presented in this study are available on request from the corresponding author.

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
