# Peer review of "Investigating the Relationships between Taste Preferences and Beverage Intake in Preadolescents"

_foods, 2023, doi:10.3390/foods12081641_

Round 1
Reviewer 1 Report
This manuscript aimed to investigate the link between taste and intake (beverage) in preadolescents (11-12 years). The study was planned thoroughly, however I have few concerns as listed below.
Authors claim that this is a first study that has been conducted
Minor comments:
1. line 45 and 64, use "," instead of"." as a unit denominator
2. line 95, include age group and gender ratio.
3. Table 3, include a space between "100ml"
Major :
1. Introduction read more like a discussion rather an intro.
2. Include a flow chart explaining study design (no. of recruits, drop outs, reason and other) for better understanding of section 2.1.
3. It is suggested to decrease the number of references older than 10 years (currently 40%) to say ≤25%.
Author Response
Thank you very much for your comments and your time you have invested in the review of your manuscript. We appreciate it very much. See our reply in the document.

Reviewer 2 Report
Dear authors
the subject is really important considering the severity of the problem of obesity in pre-adolescents. The paper can be accepted after minor revisions.
I suggest the authors to revise their results considering some of the results of the project IT-Taste, because the number of subjects analyzed is very high, even if they concern only adults. Here are some suggestions:
Dinnella C., et al., 2018, Individual Variation in PROP Status, Fungiform Papillae Density, and Responsiveness to Taste Stimuli in a Large Population Sample, Chemical Senses, 43, 9: 697–710, https://doi.org/10.1093/chemse/bjy058
Monteleone, E. et al., 2017, Exploring influences on food choice in a large population sample: the Italian Taste Project, Food Qual. Prefer. 59. doi: 10.1016/j.foodqual.2017.02.013.
Author Response
We are very grateful for you review and for your time effort. We have included the suggested references and further revised the manuscript`s results and included them also into the discussion and introduction section. Please find our reply in detail in the document.

Reviewer 3 Report
The paper is an an interesting topic that requires more research effort. The study was reasonably designed and executed - the effort working with young children is commendable.
The method section - sweetness responsiveness scale (5-point likert) seems not a conventional method to me, particularly when it is then transformed into 100 scale. Can the authors please add some justification of transforming the scale to 100?
Bitterness was tested with gLMS - so the children are instructed to use this scale. This adds to my question why it was not applied to the sweetness in the first place.
A few new references should be added to the Introduction and Discussion.
Abeywickrema, S., Ginieis, R., Oey, I., Perry, T., Keast, R. S., & Peng, M. (2023). Taste but not smell sensitivities are linked to dietary macronutrient composition. Appetite, 181, 106385.
Armitage, R. M., Iatridi, V., Vi, C. T., & Yeomans, M. R. (2023). Phenotypic differences in taste hedonics: the effects of sweet liking. Food Quality and Preference, 104845.
Author Response
Thank you very much for your review and comments. We are very grateful for your time you have invested in it. Please find our reply in detail in the document.

Round 2
Reviewer 1 Report
I do not have any further comments or suggestions on this manuscript.